# Valuation of Goat and Sheep By-Products: Challenges and Opportunities for Their Use

**DOI:** 10.3390/ani12233277

**Published:** 2022-11-24

**Authors:** Ana Rita Ribeiro de Araújo Cordeiro, Taliana Kênia Alencar Bezerra, Marta Suely Madruga

**Affiliations:** Department of Food Engineering, Technology Center, Federal University of Paraiba, Campus I, João Pessoa 58051-900, Brazil

**Keywords:** edible by-products, ethnic meat foods, flavoring, nutrition composition, protein hydrolyastes

## Abstract

**Simple Summary:**

Goat and sheep by-products are the source of many healthy and biologically active compounds that have significant potential to provide new and valuable food ingredients. However, the lack of data on these small ruminants and their undervaluation challenges the community to determine the best way to use this resource. This review highlights challenges and opportunities regarding the use of goat and sheep by-products in many different food applications. Although these by-products undoubtedly present themselves as a rich source of nutrition and allow the elaboration of many dishes, and the development of several technological applications, it is still necessary to investigate their potential use for the extraction of bioactive components, through obtaining peptides from the hydrolysis of proteins that have different biological and technological properties, in addition to the hydrolyzed use of these by-products as precursors for the production of flavorings.

**Abstract:**

Goat and sheep meat production is a challenge for the meat industry as well as for environmental management. Yet within cultures, certain by-products, such as liver, the lungs, heart, brain, spleen, blood, tail and ears, are traditionally used in the production of typical dishes for regional or local cuisine. These by-products are a rich source of lipids, proteins, essential amino acids, B-complex vitamins, and minerals. They can be effectively exploited for higher (value-added) applications, including functional foods or feed ingredients, food supplements, enzymes and other chemical products such as hydrolyzed proteins and flavorings. This review article gathers data on: (i) the production of by-products obtained from slaughter and available for processing, and (ii) potential strategies for using and applying these by-products in obtaining new value-added ingredients. Other than proteins, the review discusses other macromolecules and possible uses of these by-products in culinary dishes, as hydrolyzed enzymes, and as food additives. Even though these by-products undoubtedly present themselves as rich in nutrients, there remains an unfortunate lack of documented information on the potential use of these by-products for their bioactive components, peptides that have various biological and technological properties, and the use of hydrolyzed versions of these by-products as precursors for the production of flavorings.

## 1. Introduction

Global production of red meat has surpassed the 337 million ton mark, of which 1.82% (~6.14 million tons) is goat meat, and 2.93% (~9.88 million tons) is sheep meat [1]. The meat of small ruminants stands out among animal proteins for generally not bearing cultural and religious restrictions, and also presenting excellent nutritional quality, especially goat meat. However, it is observed that due to its sensory characteristics, goat meat is not a favorite in Western countries, and is rejected by some consumers. However, we note that the demand for this meat in developing countries, especially in semi-arid regions, goes beyond production. These areas account for more than 90% of the world’s goat herd with 780 million heads [1]. Improving production standards and the availability of specialized products prepared from these meats may well contribute to popularized consumption [2].

Increases in goat/sheep meat production are accompanied by increases in edible by-products, generated not only from slaughter, but also from meat processing [3]. It is estimated that about 3 million tons of goat and sheep by-products were generated in 2018, which represents, on average, 15 to 20% of the total live weight of the animals [1,4].

If not properly treated, slaughter by-products pose both environmental and economic problems for meat processing plants, yet various industrial processes to develop meat products, hydrolyzed proteins, and flavorings are being proposed to treat these by-products, usually discarded or seen as products of lower value [5]. Studies have shown that meat by-products are a valuable raw material since they bring excellent nutritional value and are rich in lipids and proteins. This justifies a wide variety of applications for originating new ingredients and food products [6,7].

Goat and sheep meat by-products are used in various traditional preparations of typical regional dishes, as reported by [8,9,10]; and in the development of processed products, such as pâtés [11,12,13,14,15], sausage [16,17], and chorizo [18], among others.

In general, optimizing the use of slaughter by-products reduces environmental impacts by enabling the production of new products and ingredients. Technologies that recover proteins from meat by-products add value to them [19,20]. The use of goat and sheep waste to obtain flavorings through enzymatic hydrolysis [21,22], thickening agents, emulsifiers, foaming agents, and stabilizing agents [7] has recently drawn attention [4].

Goat and sheep slaughter by-products make an excellent study, since the literature has demonstrated their nutritional importance, in addition to their many applications-use for food, development of meat products, and production of hydrolyzed proteins, all of which are associated with market value [11,18,23].

This review, therefore, aims to discuss edible by-products from the slaughter of small ruminants, presenting aspects involving the use of edible goat and sheep by-products (especially organs and viscera), bringing value, and providing information to promote and encourage their use. It will analyze production issues and the challenges and concerns that the use of these by-products requires. In addition, this review will describe the nutritional quality of these by-products in their use as food, as well as the potential of hydrolyzed proteins in technological, bioactive, and flavoring agent applications. Through a broad scientific approach, this review will contribute to understanding the use of goat and sheep by-products within the greater food matrix, knowledge that has not been previously addressed in the literature.

## 2. Aspects Involved in the Production of Goat and Sheep By-Products

The production and consumption of goat and sheep products have seen a global increase over the last 60 years. The global production of goat meat, in 2018, reached figures of 6 million tons, while sheep meat production reached a total of 9.78 million tons [1]. Although the production of meat from small ruminants is less than the production of other meats (beef, poultry, and pork) [1], there is a growing demand in consumption that needs to be met. The authors of [24,25] point out the many challenges in small ruminant meat production and promotion, these include genetic (herd) improvement, producer training, innovation of slaughter and processing facilities, treatment of by-products, analysis of markets and marketing channels, consumer education towards the consumption of goat and sheep meat, and regional preferences as well.

The by-products generated during slaughter account for approximately 15 to 20% of the total live weight (of goats), and up to 18% (of sheep) (Figure 1). These percentages, in 2018, respectively, generated around 1.17 and 1.72 million tons of goat and sheep by-products [1,4].

Edible by-products: liver, lung, heart, tongue, kidney, brain, spleen, blood, tail, and ears represent an excellent source of income for slaughterhouses and are relevant sources of micronutrients such as essential amino acids, minerals, and vitamins [4,26,27,28].

Consumption of the edible by-products of small ruminants remains diverse, being appreciated differently according to tradition, culture, heritage [8,9,27,29], and by region or country [10,30]. In France and Ireland, for example, consumption of by-products is an important part of meat consumption (~8 kg/person), while mocotó, brain, and other by-products are considered delicacies in Asia, but in most developing countries they are not consumed [1].

With the rapid growth of the economy in many countries in Asia, the trade in animal by-products is growing rapidly. In China, liver and kidneys are traded at prices five times higher than in the United States. In Brazil, the market of goat and sheep by-products still shows low levels of organization, which reflect on the quality of the products, the lack of supply stability, the productivity levels, and mainly on the informality of distribution. The informal trade of these by-products can be observed in most of the country, with greater prominence in the northeastern semi-arid region. However, in the traditional markets of the northeast and south regions of Brazil, the consumption of goat and sheep products has increased significantly. In addition, boutiques and restaurants specializing in receiving these by-products have attracted many consumers in the country’s tourist and coastal cities, where by-products with a high-quality standards are offered [31].

In the United States, everything that is produced from an animal, except meat, is classified as a by-product which then divides into edible and inedible. Head trimmings and fat from the rumen and stomach are included in the definition of edible by-products [32]. In the United Kingdom, this listed segregation takes place between red (liver, lungs, tongue, head, etc.), and white (fat) by-products, the viscera and bladder, the rumen, and legs. Since the outbreak of bovine spongiform encephalopathy, the spinal cord and brain have not been re-included and mammalian blood, lungs, stomach, intestines, feet, and testicles are items that cannot be used if uncooked [5,32].

Since the early 2000s, new European and US regulations have opened new paths and markets for the use and exploitation of animal by-products, which for a long time were undervalued. An example of this is European Directive (EC, 2008) which prevents waste, and promotes both its recovery and use. This directive defines as a by-product, “any substance or object resulting from processing, which was not the main object of production, and which at the same time, must not be wasted” [3].

In Brazil, industrial and sanitary inspection regulations also consider the organs and viscera of butchery animals, as being used in human food, called “miúdos” (brain, tongue, heart, liver, kidneys, rumen, reticulum, omasum), and include hoofs and tails [33]. According to the official nomenclature of meat products in Brazil, the liver, heart, kidneys, tongue, stomach, lungs, spleen, and thymus are offal whether of goat or sheep [34]. Brazilian legislation considers it mandatory that by-products be submitted to inspection before being released, to then continue to handling, cleaning, and preparation, either for better presentation and/or further treatment [33].

In 1986, due to the occurrence of bovine spongiform encephalopathy (BSE) in Europe, certain implications for safety in meat and by-product consumption emerged. In 2001, a large group of animal proteins used in animal feeds, including meat by-products, was banned by the European Union after being implicated with BSE [35]. European regulations do not currently prohibit the use of by-products in animal feed, but public health surveillance requires that food safety related to the use of meat by-products in such products be continually evaluated. Yet, in the UK, for example in food use, the spinal cord and brain continue to be banned since the BSE outbreak. Further, in many countries, such mechanically separated meat suffers from poor consumer perception, due also to health concerns about bovine spongiform encephalopathy contamination [32,35].

## 3. Nutritional Composition of Goat and Sheep Edible By-Products

### 3.1. Goat and Sheep Meat

Though nutritional qualities vary between species (due to many factors) protein, minerals, and vitamins are generally consistent [36]. We note protein (respectively) for goats and sheep: goat meat (21.1%), sheep meat (24.0%), moisture contents (76.7% and 74%), and ash (0.92% and 1.15%). Goat meat stands out for having much lower percentages of fat, with values of <3%, while sheep meat reaches levels >8%, which is roughly three times higher [37]. The meat protein of these small ruminants possesses all of the essential amino acids, with low caloric values, and reduced fat and cholesterol content, thus meeting the needs of consumers focused on alternative consumption of healthier foods. [36,38]. Comparisons between the amino acid profile of an ideal protein and that of goat and sheep meat (Table 1 and Table 2) reveal essential amino acid percentages of greater than 80%, excepting only phenylalanine (60% and 57%), isoleucine (77% and 70%), and valine (77 and 78%).

Goat and sheep meat are good sources of many minerals, including calcium (Ca), phosphorus (P), sodium (Na), and magnesium (Mg), (the most common macro-elements), as well as iron (Fe), copper (Cu), selenium (Se), zinc (Zn), manganese (Mn), cobalt (Co), nickel (Ni), vanadium (V), lead (Pb), and cadmium (Cd), which are considered important trace elements. The contents of Ca, Fe, and P found in meat from goats and sheep vary in concentrations: P = 141 to 271 mg 100 g^−1^ for goats and 176 to 215 mg 100 g^−1^ for sheep; Ca = 2.3 to 9.6 mg 100 g^−1^ for goats and 5 to 7 mg 100 g^−1^ for sheep, and Fe = 2.8 to 11 mg 100 g^−1^ goats and 1 to 2.2 mg 100 g^−1^ for sheep. Of these minerals, Fe is found in higher concentrations in goat meat, reaching levels three times higher than other meats. The nutritional importance of iron for brain function, physical activity, and oxygen transport naturally invite greater consumption of goat meat [35,36]. Goat meat is also an important source of vitamins B1 or thiamine (0.11 mg 100 g^−1^), B2 or riboflavin (0.49 mg 100 g^−1^), and B12 or cyanocobalamin (1.13 mg 100 g^−1^). Sheep meat is another source of these vitamins, with similar values of 0.105 mg 100 g^−1^ for B1, 0.28 mg 100 g^−1^ for B2, and 0.0026 mg 100 g^−1^ for B12. Goat meat itself presents higher riboflavin contents than other meats, and all of these vitamins are essential for protein and carbohydrate metabolism, fighting anemia, and promoting growth.

### 3.2. Goats and Sheep: Edible By-Products

Edible by-products from the slaughter of goats and sheep are significant sources of essential nutrients, and as does the meat itself, they enjoy comparable nutritional values. The scientific data on the nutritional value of these by-products are relatively scarce, with values available on the internet in the limited form of nutritional food tables; these being much fewer in number for goat by-products. In [39], the authors have described the composition and nutritional value of raw and cooked edible sheep by-products together with certain other species. Ref. [27] has published macro- and micronutrient compositions for by-products of various species, however, no data have been presented in either review for goat by-products. It is noteworthy, however, that the results reported include chemical analysis, proximate compositions, cholesterol, fatty acid profiles, and sheep by-product minerals, especially from the brain, heart, liver, and tongue [3,27,28,39,40,41].

Other studies have addressed the characterization of amino acids, fat, cholesterol, and moisture in goat and sheep by-products. The amino acid profiles found in bovine, swine, and ovine meat organs demonstrate that, in general, leucine, valine, methionine, and threonine are found in higher concentrations in the liver than in other organs. Amino acid concentrations tend to be lower in the lungs and spleen, however, the higher content of tryptophan found in the pancreas is highlighted [42].

The chemical characteristics found in by-products: brain, heart, kidneys, liver, (and the tongue, in Iraq) for sheep and cattle when roasted or more conventionally cooked have also been studied [43]. In raw and grilled samples, moisture, fat, and cholesterol varied significantly. Grilled and cooked samples present higher fat and cholesterol, possibly due to the lower moisture found in these samples.

Chemical characterization (and effects) of a calcium-rich diet (based on) liver, kidney, and heart samples from goats: *Longissimus dorsi* (LD), and *Biceps femoris* (BF), was performed by [44]. The results revealed that fat and cholesterol are influenced by breed, tissue type, and dietary calcium. Lower cholesterol contents were found in goat hearts and livers (at 167.5 and 214.2 mg/100 g tissue), yet higher % cholesterol contents were found in kidneys (276.7 mg/100 g tissue).

Table 1 and Table 2 compile in detail data on proximate composition, amino acids, lipids, minerals, and vitamins for both goat and sheep by-products; below, we discuss this information while focusing on the values obtained in published articles.

### 3.3. Centesimal Composition

Moisture, varying from 66 to 81% is the predominant component of edible meat by-products in small ruminants. Interestingly, the highest moisture level is found in goat casing–intestinal sub-mucosa (80.9%). The lowest moisture levels are found in sheep tongue (66.6%). The lungs, heart, and brain also present high humidity levels, ranging from 76.7% to 80.3% (see Table 1 and Table 2).

Protein contents (with small exceptions), for goat and sheep by-products, stand out, with percentages that vary in general from 15 to 20% (Table 1 and Table 2). This is probably due to the contents of other components such as moisture and lipids. The liver, lungs, heart, and spleen present the highest protein concentrations, with values ranging from 15 to 22%. The lowest protein content is detected in the brain.

For macro constituents, fat percentages in goat and sheep edible by-products present a greater variability (<2% to >17%). It is known that small ruminants tend to deposit high levels of visceral fat, especially goats; this explains the variability [45]. However, fat content is normally low, usually < 5% in the lungs, spleen, and kidneys of both sheep and other species. These values are reported by [27]. Higher values of fat are found in the brain, heart, pancreas, and tongue (>5.68). Fat content in these by-products influences physical, chemical, sensory, and nutritional properties [46]. The liver and kidney organs contain negligible amounts of carbohydrates [39]; while [27] reported 5% carbohydrate values for the liver, and lower values for other organs.

There are small differences in moisture content, total fat, and cholesterol content in goat liver, kidneys, heart, and meat. Noting that there are significant differences depending on the breed were reported [44]. The nutritional value of lamb organs in two South African breeds, Dorper and Merino, is reported by [47]. However, the brain, kidney, liver, lung, and spleen organs were found to be rich in cholesterol in either breed, and both should be restricted in human diets.

Calculating by-product caloric values, it was observed that all those with low fat and carbohydrate content provide values lower than 100 kcal/100 g (Table 1 and Table 2). Higher caloric intakes are observed for the tongue, brain, and liver (>100 and >230 kcal/100 g). Caloric values in edible sheep by-products ranging from 95 to 150 kcal/100 g have been reported [27].

Edible by-products from goats and sheep present high cholesterol contents (112 to 1350 mg/100 g), some, such as the brain, present levels of up to twenty times higher than those of the meat (50 to 70 mg/100 g). The kidneys, liver, lungs, pancreas, and spleen present cholesterol levels ranging from 250 to 371 mg, three to five times higher than that of lean meat. The tongue presents approximately 135 mg of cholesterol, and the heart presents roughly 156 mg of cholesterol/100 g. According to [27] cholesterol content does not strongly depend on the fat contents, since it remains a constituent of cell membranes, necessary for functionality. Cholesterol content is more dependent on cell size (surface area) than on fat content. In goats of various breeds, ref. [44] reported only small differences in liver, kidney, and heart cholesterol.

### 3.4. Amino Acids

Between by-products and species (Table 1 and Table 2), amino acid composition presents slight variation in values; essential and non-essential amino acid composition remain reasonably constant. Essential amino acids in goat and sheep by-products are rarely described in the literature. The amino acid content of edible meat by-products of lamb and other mammalian species was revised [39]. The amino acid profile of goat viscera confirmed that these by-products are an excellent alternative for the production of new food products or functional ingredients [4]. The amino acid profile has also been reported in traditional products such as “sarapatel” [48], and “buchada” [49], products made with goat and sheep by-products and viscera.

Among essential amino acids, lysine, phenylalanine, and methionine stand out in goat and sheep by-products. Arginine and cystine are non-essential amino acids and yet also present higher concentrations in these by-products [7].

In edible goat and sheep by-products, isoleucine content is higher by 35%. For kidneys, tripe, and sheep tongue, values higher than 50% are obtained. The lowest concentrations of valine in the brain (59.86%) and tongue (51.71%) occur in sheep, other by-products present percentages above 60% (Table 1 and Table 2).

In meat by-products, essential amino acids are a very important group of nutrients, which do not suffer losses, either from cooking and processing, or via secondary degradation Maillard reactions (low carbohydrate contents), racemization, or cross-linking reactions [26].

### 3.5. Fatty Acids

The fatty acid composition of goat and sheep by-products has been well studied. Some investigations [39,47,48,49,50,51,52] present the composition of fatty acids in organs and viscera for both species.

In the fatty acid profiles of these by-products, there are higher concentrations of saturated fatty acids, especially palmitic and stearic acids. Monounsaturated (oleic) and polyunsaturated (linoleic and arachidonic) fatty acids (in tongue and brain by-products) enjoy higher concentrations. Lower values of saturated and monounsaturated fatty acids are found in the lungs, kidneys, and spleen, which also present reduced fat contents (Table 1 and Table 2).

The monounsaturated and polyunsaturated fatty acid sum (Table 1 and Table 2) is greater than saturated fatty acid (SFA) content, indicating that the SFA occurs in percentages of ≤50%. Values above 45% are observed for the monounsaturated fatty acids present in goat by-products, the same type of healthy fat found in olive oil. Monounsaturated fats are thought to lower total cholesterol and reduce the risk of heart disease [53]. From 32% to 57%, respectively, of the saturated fat from sheep and goat by-products is stearic acid, which is neutral toward total cholesterol levels in humans [54].

Oleic, octadecanoic, hexadecanoic, and linoleic acids are the principal fatty acid profile components in lamb offal [39]. Further, it has been noted that the average percentages of hexadecanoic, octadecanoic, and oleic fatty acids in goat by-products are similar to those reported for goat meat [53,54,55].

FA compositions in Longissimus dorsi (LD) and Biceps femoris (BF) of the Alpina and Nubiana goat breeds for the liver, heart, and kidneys were evaluated by [50]. They noted that all organ samples presented higher polyunsaturated fatty acids (PUFA) concentrations than the muscles (*p* < 0.001). The organs also contained significantly (*p* < 0.01) higher percentages of saturated fatty acids (SFA). The fatty acid composition of “buchada”, a goat meat by-product, mainly contains stearic acid, oleic acid, palmitic acid, and low unsaturated fatty acid levels [48,49,51].

Fatty acid profiles differ between organs and breeds of sheep; however, fewer differences are observed for total SFAs and MUFAs [47]. Yet Dorper heart (1.8%), presents lower total PUFA than Merino heart (7.3%), *p* < 0.05. All organs (with the exception of the tongue, heart, and stomach) present favorable P:S ratios. The brain, lungs, and testicles of Dorper and Merino sheep present favorable n-6/n-3 ratios. Barbari goats present superiority in the PUFA/SFA and n6/n3 ratios for the viscera and *Longissimus lumborum* (muscle), over goat muscle [52].

### 3.6. Minerals

Information on mineral profiles for edible goat and sheep by-products is scarce, noting that the by-products have been little analyzed, and data are mostly for the liver and kidneys. in nutritional terms (micro-mineral data in by-products of differing species, including sheep), the concentration of minerals is lower in organs and viscera as compared to muscle [41]. In the composition of minerals in raw and cooked sheep viscera and muscle, it was observed that (as a result of concentration) cooking generally tends to increase the levels of minerals such as calcium, magnesium, phosphorus, zinc, copper, and manganese [39].

Differences (*p* < 0.05) between meat and by-products for *Longissimus lumborum* muscle and (*Capra hircus*)-{Barbari kids} as to the mineral profiles they have been observed [52] The kidneys presented the highest sodium content (202.39 mg/100 g), while the potassium content was higher in the testes (362.61 mg/100 g). Copper, iron, and zinc were found in, respectively, higher concentrations in the liver (6.97 mg/100 g), spleen (31.16 mg/100 g), and muscle (4.15 mg/100 g). Thus, edible by-products present high concentrations of essential minerals and can provide an appreciable amount of their recommended daily allowances.

Potassium, phosphorus, and sodium were detected in higher proportions in the by-products of goats and sheep (Table 1 and Table 2). Phosphorus and potassium levels were highest in the lung, brain, spleen, kidneys, tongue, and liver. Sodium is prominent in the tongue, kidneys, and brain, with values higher than in the meat of the respective species.

Breaking things down by by-product, the kidneys present higher proportions of calcium, potassium, selenium, and sodium. The heart stands out for its arsenic and potassium contents, and the pancreas for calcium and magnesium. The spleen presents large amounts of iron, magnesium, and potassium. The liver is rich in copper, phosphorus, and zinc, and the tongue is rich in magnesium.

Iron content is more abundant in the spleen and varied from 51.4 mg in sheep to 53.1 mg in goats. The kidneys, lungs, and liver presented iron contents ranging from 6.11 mg to 8.58 mg. Magnesium is found in the highest percentages in the tongue (19.4 to 25.5 mg, respectively), though being the lowest in the lungs (11.5 to 14 mg). Calcium content is highest in the pancreas, ranging from 14.85 mg for sheep to 20.87 mg for goats. Most by-products present high selenium values, especially the kidneys (127 to 142 mg), followed by the spleen (48.5 to 64.5 mg), and liver (43.4 to 48.1 mg). Zinc content is highest in the liver, followed by the tongue, spleen, and kidneys.

### 3.7. Vitamins

Lipo and water-soluble vitamins in goat and sheep by-products are rarely reported. Most edible by-products (heart, kidney, liver, pancreas) are excellent sources of B-complex vitamins. Vitamin A is usually found in higher concentrations in the viscera compared to meat. Vitamin C is prominent in the kidney, liver, and pancreas.

The vitamin composition in raw sheep by-products was listed [28,39], noting reasonable losses in vitamin content during the cooking process.

For vitamins (Table 1 and Table 2), it can be observed that thiamine and riboflavin values are similar for both sheep and goats, they are essential for growth, carbohydrate and protein metabolism, and for fighting anemia [45]. The liver, tongue, and lung contain the highest amounts of thiamine and riboflavin while the other by-products contain <0.15 mg of thiamine and <0.22 mg of riboflavin.

Thiamine, an essential vitamin for the proper functioning of nerve cells and the brain, when subjected to high temperatures, undergoes thermal degradation and can release volatile compounds (responsible for the characteristic “flesh” odor and taste). Thiamine is a good precursor of meat aroma during the cooking process [56].

The liver is highlighted by its niacin content, being considered the best source. Except for the lungs and kidneys, the other by-products present similar vitamin B6 values in both species, with values ranging from 0.16 to 0.51 mg/100 g. Pantothenic acid is highest in the lungs, spleen, and tongue. The lungs and tongue are the best sources of vitamin B9 in both ruminants (goats and sheep).

## 4. Cultural and Culinary Use of Goats and Sheep Edible By-Products

In general, the consumption of offal is part of the human menu, daily intake, (in different ways and different countries), offers a variety of foods that are nutritionally attractive. Edible organs are highly prized in Southeast Asia and Africa, and demand is low in the US, yet variable in Australia [57].

For example, the heart is often used as table meat, often roasted or sautéed, and it can also be used (as an ingredient) in processed meat [32]. Kidneys can be boiled, grilled, braised, and fried [58], and liver, the most consumed, can be used in sausage and pate [11].

In the United States and Europe, lamb liver is well accepted due to its light flavor and texture [59]. Lamb lungs are used in certain sausage types and other processed meats [25,32].

Intestines find their first application as sausage casing, but in some countries, they are also used as food after being boiled [58]. Ears are smoked and salted or cooked with the feet. The spleen is usually eaten fried or used to prepare chorizo [32].

Goat and sheep by-products, offal, organs, and blood stand out for being part of differing cultural identities, playing a very important role and constituting elements of the gastronomic and nutritional heritage, which have been consumed for centuries. Many of these products, called “ethnic meat foods” are unknown to the general public, they are based on foods in developing country populations, with peculiar sensory qualities, and yet provide high nutritional value and even nutraceutical characteristics [8].

“Ethnic foods” are uniquely regional and community specific. Some are exotic and rare, though often including meat, offal, organs, and blood. Some “ethnic meat products”, such as sausages, and dried or smoked meats are sold in local markets and contribute to the local economy [9].

The preparation of most “ethnic goat/sheep meat foods” presents its own characteristics and typical organoleptic aspects. The aspects are a result of the addition of herbs and spices during preparation, the condiments result in important antimicrobial activity and improve the microbiological quality of the final products [8].

Articles focusing on the quality of “ethnic goat/sheep meat foods” note that most “ethnic meat foods” (traditional products) are produced only with meat from goats and sheep, such as “Cecina de goat”, “Cecina de castron”, or “Violin di capra”, etc., were published [8,9,10]. However, many other “ethnic meat foods” are made from viscera, blood, and organs, etc. Unlike traditional meat products from swine, cattle, and poultry, publications on microbiological and nutritional aspects of these meat products are limited.

Certain “Ethnic Meat Foods” produced with goat/sheep by-products are listed in Table 3. Among them are minced viscera and blood/“Buchada de bode”, and “Sarapatel” in Brazil; blood-fermented sausages/“Blóðmör”, and sheep liver/“Lifrarpylsa” typical of Iceland; lamb sausage prepared with by-products/”Osbana” consumed in Algeria, Tunisia, Libya, and Morocco; and “Kourdass”, traditional in North Africa and the Mediterranean [8,10,48,49].

“Ethnic meat foods” have been reported in various articles, “Buchada” and “Sarapatel” [65], which are typical dishes of the Brazilian Northeast, are sold in restaurants, bars, street markets, or even prepared at home. They contain the heart, lungs, liver, intestine, stomach, blood, tomato, pepper, coriander, onion, garlic, and other spices [48,49]. These are the object of studies reported by [48,49,51,62,64,66,67] who affirm the feasibility of using goat and sheep by-products to prepare these meat products and gain more than 55% in relation to the simple carcass value. They also augment the nutritional value of the final product, mainly due to the high protein contents, satisfactory fatty acid profile, and essential amino acids. Ethnic meat foods carry both microbiological and social importance.

Microbiological conditions in pre-cooked goat buchada were evaluated, noted inadequate hygienic-sanitary practices, and the need to implement better manufacturing practices, to guarantee products with a longer shelf life and higher quality standards [66]. The microbiological quality of buchada and sarapatel were evaluated [48,49], sold under different conditions—street markets, butchers, and in supermarkets; under refrigeration, frozen, or at room temperature—they noted the absence of *Salmonella* spp. and *Listeria monocytogenes*, and high total and thermo-tolerant coliform counts. Sulfite-reducing Clostridium and coagulase-positive Staphylococcus were also detected, indicating that unsatisfactory microbiological conditions had affected sanitary aspects during the processing and selling of these products.

Considering the high counts reported in previous studies, ref. [64] studied the effect of citric and malic acids on sheep buchada food safety, reporting that the addition of these organic acids decreased pH and moisture in the samples, and increased acidity, which can be determining factors (barriers) in microbial growth, yet without negatively affecting sensory attributes.

The profile of fatty components (fat, phospholipids, cholesterol, and fatty acids) was analyzed in five formulations of goat buchada made with viscera and blood, they observed higher fatty component profiles than in goat meat. The main fatty acids were stearic acid, oleic acid, and palmitic acid, with low levels of unsaturated fatty acids [51].

Sheep buchada formulations exhibited higher protein percentages (24.75–27.04%) and lower fat contents (<5%) [64]. However [48,49,62], when quantifying the chemical composition of buchada/sarapatel, observed lower protein (12 to 18%) and higher fat percentages (>5%). These authors demonstrated that buchada/sarapatel is a nutritious food, with high protein contents, fat (9%), and important fatty acid and amino acid profiles, being thus an alternative to the use of by-products such as blood and viscera.

Articles have also reported on the development of meat products processed with by-products (viscera, blood, organs) from such small ruminants, but not as part of the gastronomic heritage, such as smoked sausage [16,17,18] and pates [11,12,13,15].

The preparation of smoked sausages made with blood, viscera, and traces of goat meat was evaluated [16,17]. It was demonstrated that these products are rich in proteins of high biological value, essential amino acids, fatty acids, and iron, obtaining high sensory acceptance (over 80%). Smoked sausage presents better flavor and aroma sensory characteristics and stands out from other formulations (for presenting high iron contents while being preferred).

The use of goat blood, liver, and meat trimmings in goat pate preparation have revealed a better added-value product. The main nutritional characteristics obtained from these formulations were high protein and iron contents, yet physicochemical properties must always meet legal requirements as well [11]. The nutritional quality of a pate made with blood, liver, and sheep meat trimmings was evaluated by [12], revealing essential amino acids, proteins (15.1%), iron, and linoleic acid (16.68%), with good sensory acceptance that the pâté may be used as a valuable sheep by-product.

Sheep and goat meat pâtés with different fat proportions (swine belly or oil) studied by [15], presented low-fat content (9.7–18.2%), and high protein values (22–24%). The products obtained possessed fats, which according to dietary recommendations were in balance, demonstrating the value-added potential.

The different results obtained in the studies mentioned above serve to demonstrate that the use of by-products from the slaughter of goats and sheep is viable for obtaining new food products with desirable nutritional qualities. The use of these components also makes it possible to expand the variety of products that can be offered, generating more income and promoting the development of the production chain.

## 5. Application of Goat and Sheep By-Products to Obtain New Ingredients

Given the characteristic nutritional potential of such by-products and using an innovative approach, biotechnological processes can be applied. The use of by-products from small ruminants goes beyond the preparation of traditional dishes and processed meat products for human consumption. Better use of by-products makes it possible to reduce their environmental impacts while enabling the production of new ingredients and products. Different processes and technological methods may be applied to by-products to obtain products with various functionalities.

Due to their high composition in proteins and differentiated amino acid profiles, such by-products have been widely used in their hydrolyzed form [19,20]. The hydrolysis process consists of breaking the peptide bond of proteins, resulting in the release of peptides of different molecular sizes, distinct amino acid sequencing, and free amino acids [63].

Hydrolysis can be carried out using enzymatic, chemical, or thermal methods. The chemical and thermal methods are considered difficult to control and result in products with lower nutritional and/or bioactive quality. Methods based on the use of enzymes are preferentially chosen in the food and pharmaceutical industries. Enzymatic methods do not use organic solvents or chemical products that present toxicity, and yet are also fast and controllable; advantages over other methods [68,69].

The properties assumed by the final hydrolysate will fundamentally depend on the degree of hydrolysis (DH). DH is defined as the percentage of the total hydrolyzed peptide bonds at the end of the hydrolytic process; thus, if the protein is fully transformed into free amino acids, DH = 100%. In general, a high DH means that low molecular mass peptides have been formed (in addition to high amounts of free amino acids). This increases protein nitrogen recovery yields, provides better solubility, and probably a more bitter taste [20,21].

Studies have been performed (specifically hydrolyzing by-products from the slaughter of goats and sheep) and are shown in Table 4. In [7], authors observed with chemical characterization the potential for goat viscera in the use of hydrolyzed proteins, concluding that the protein profile found in these by-products (with a majority presence of glutamic acid, aspartic acid, lysine, and glycine) indicates potential for use as a flavoring/flavor enhancer. The hydrolysates possessed significant technological–functional properties, such as solubility, oil retention capacity, emulsifying properties, and antioxidant activity.

Protein hydrolysis process conditions for sheep viscera (stomach and intestine) were studied by [70], who hoped to achieve the maximum antioxidant activity of the hydrolysate. The hydrolysate obtained presented high protein (83.78%) and low fat (0.34%) contents, as well as exhibiting high DPPH antioxidant activity (68.21%), thus making it possible for use as a natural antioxidant with high nutritional value.

Another study involving the production of hydrolyzed proteins prepared from sheep visceral mass (stomach, and large and small intestines), ref. [71] concluded that these by-products present considerable potential for hydrolysate production and exhibit high concentrations of certain essential amino acids.

Further, the protein hydrolysate has been characterized by the presence of low molecular weight peptides, which suggests the presence of peptide bioactivity. The relatively high values of the corrected amino acid score for protein digestibility and digestibility in vitro indicate some appropriate use for such hydrolysates as a nutritional supplement in many foods [71].

Using sheep plasma as raw material, three peptides were isolated and characterized from different fractions, using protein hydrolysate with alcalase. The hydrolysate obtained exhibited considerable Ferris-reduced antioxidant power (FRAP) activity and stable free radical scavenging activity 1,1-diphenyl-2-picrylhydrazyl (DPPH). The sequences found included amino acids that were previously listed as major contributors to the peptide’s antioxidant properties. Thus, sheep plasma is considered a good source of antioxidant peptides, enabling its use as a useful food additive or as bioactive material, a new opportunity for using animal blood by-products [76].

The effects of adding powdered sheep offal protein hydrolysate on the oxidative stability of two matrices (soybean oil and chicken sausage) was evaluated by [73]. The research revealed that a concentration of 700 ppm of viscera hydrolysate was able to control oxidation in soybean oil, in addition to imparting considerable stability effects in chicken sausage. The hydrolysate under study can be used as an enriching additive or as a natural antioxidant in lipid-containing oils or products such as sausage.

The use of sheep abomasum hydrolyzed proteins to obtain antioxidant peptides demonstrates that these purified peptides are natural antioxidants, suggesting their use in food additives and pharmaceutical products [74].

The enzymatic hydrolysis process is able to improve proteins in terms of their functional, sensory, and physicochemical characteristics; permitting their use as supplements, and obtaining new food products; to provide functional properties, and reduce allergenic potential, while producing flavor components [77].

In addition to using the hydrolysis process in by-products, extraction of collagen from by-products in differing species is promising and is currently under study [78,79,80].

Collagen from sheep slaughter by-products was extracted and characterized, obtaining respective yields of 18.0% and 12.5% for lamb and ewe by-products [81]. The extracted collagens possess important technological properties: (acid solubility, foam formation, and emulsifying power) which allow food product applications, adding value, and providing a sustainable destination for these slaughter by-products.

Production of flavor molecules from goat by-product hydrolysates, emphasizing the thermal action during processing was studied [22]. They observed that the application of two heat treatments resulted in the production of hydrolyzed proteins with different volatile profiles. The hydrolysate obtained without prior heat treatment, but with autoclaving, presented a higher concentration of precursor molecules involved in the meat flavor reactions, contributing to the generation of heterocyclic compounds, and impacting the quality of goat hydrolysate aroma.

### Flavors Obtained from Goat and Sheep By-Products

Flavor is a sensory impression perceived through a combination of taste and odor, in the consumer’s purchase decision, it is identified as one of the most important characteristics of goat and sheep meat [82]. Unlike flavor components (which are generally non-volatile), odor components are comprised of a combination of active compounds that may be engineered through volatile organic molecules. The complex aroma system is related to interactions between volatile compounds and other elements present in the food [83].

Soluble, low molecular weight compounds, and fats are the main precursors conferring flavor to cooked meats [84]. The Maillard reaction, lipid oxidation, and vitamin degradation are the main reactions involved in triggering the aroma of cooked meat. In addition to these, animal factors such as feed, sex, age, and breed (with cooking conditions and aging) contribute to aroma [82].

The quality attributed to the aroma of a protein hydrolysate will also depend on the initial quality of the raw material, which influences the odor and taste. The degree of hydrolysis will affect sensory perception and is influenced by the type and amount of enzyme used, the pH, and temperature of the reaction, as well as the water: substrate ratio [21,84]. Heat is responsible for volatile compound increases, and not only hydrolyzes the protein, but also affects peptide contractions and types [85]. To obtain a functional peptide, proper hydrolysis conditions must be met [19,62,86,87].

During the process of protein hydrolysis, peptides, free amino acids, and volatile compounds are formed, which together present a characteristic aroma. This allows for specific applications of a protein hydrolysate as a food flavoring [7,88]. By controlling the degree of hydrolysis under thermal reaction, it is possible to obtain different precursor constitutions, and generate a considerable difference in flavor due to the differing degrees of volatile formation. Thermal reactions are important to flavor outcomes [21].

Hydrolyzed proteins obtained from by-products are used to produce flavor components [20,21,73]. Information is already available on the *flavor* of chicken, beef, pork, and fish meat. However, published data on the *flavor* of goats and sheep is still scarce [37,84,89].

The volatile compounds present in goat meat were studied by [89], who identified and quantified 203 compounds. Phenols (4-methylphenol and 2,6-dimethylphenol) were extracted and reported to contribute to mutton and goat aromas. A large number of Maillard-derived compounds, such as pyrazines, pyrroles, pyridines, and alkyl sulfonates were also found. Compounds such as 1,2-methyltridecanal, (E,E)-2,4-decadienal, 3-(methylthio)propanal, dimethyl trisulfide, and an interesting series of C2 to C5 alkylformylcyclopentenes were identified for the first time in the volatile profile of goat meat.

In a later study, [84] determined the concentrations of certain water-soluble precursors present in raw and cooked goat meat aromas. The biggest losses during the cooking process were fructose, glucose, inosine 5-monophosphate, and cysteine, which seem to be involved in the formation of goat aroma. Fructose and glycine were present in high concentrations in goat meat, in addition to other sugars, ribonucleotides, creatinine, and free amino acids.

The study of flavor molecules (obtained from goat viscera hydrolysates), demonstrated how much the thermal process quantitatively and qualitatively influences the production of these molecules [22]. The results allow us to infer that the autoclaving process provides a significant increase in the concentration of free amino acids, such as glutamic acid, alanine, arginine, valine, leucine, lysine, and phenylalanine, as well as a reduction in glucose content, and an increase in maltose content. Further, the presence of oleic and linoleic fatty acids, which play an important role in the formation of volatile compounds characteristic of meat aroma, were present in the hydrolysates. Indicating that the autoclaving process favors the production of pyrazines and other heterocyclic compounds.

Mass spectrometry (CG-MS), olfactometry (CG-O), and descriptive sensory analysis have been used to describe changes in the aroma characteristics of sheep flavorings prepared from sheep bone hydrolyzed proteins with different degrees of hydrolysis. The results revealed that the degree of hydrolysis is an important indicator in preparing these flavorings, with a significant effect on sensory attributes (roasted sheep). It was also possible to detect 36 key compounds representing the mutton flavor. Hydrolysates with a hydrolysis degree range of 25.92–30.89% presented more odor-active compounds through the thermal reactions and were indicated for use as a desirable precursor for the production of mutton flavorings [21].

Fatty acids derived from goat and sheep products are also volatile compound precursors, and thus the use of by-products from these animals in obtaining hydrolyzed proteins increases their applicability in food technology for both flavorings and functional ingredients [49]. These properties favor using them in both animal feed and human food [63].

The hypothesis that certain fatty acids with branched chains with a methyl group, present in the subcutaneous fat of lambs and goats, would be components directly responsible for characteristic “sheep” and “goat” odors was initially defended by [90]. This thesis was reinforced through sensory analysis, in which the “sheep” odor was found to be related to the presence of 4-methyl octanoic acid (4-Me-8:0) and 4-methyl nonanoic (4-Me-9:0). Following the same line of reasoning, ref. [89] studying the aromatic properties and threshold values of branched-chain fatty acids, reported that the 4-ethyl octanoic fatty acid (4-Et-8:0) presented the lowest threshold value of the twenty and three acids analyzed, also noting that fatty acids containing branches in the “4” position had characteristic “goat and sheep” odors, as well as fatty acids containing chains with eight carbon atoms.

Current studies have shown that “mutton flavor” comes particularly from subcutaneous adipose tissue, and that 4-Methyl-8:0, 4-Ethyl-8:0, and 4-Methyl-9:0 are the three main branched-chain fatty acids responsible for conferring the species-specific flavor in sheep [90,91]. Based on this, studies on the taste of goats and sheep, have sought to score consumers’ taste intensity preferences [92]. Another study suggested sensory testing and higher scores for the attributes “mutton” and “goat” in samples enriched with 4-Me-8:0 and 4-Me-9:0 fatty acids, against those without the addition of 4-Me-8:0 [90].

Further research revealed that in male goats, the 4-Et 8:0 fatty acid was specifically associated with goat aroma [93], and the same fatty acid was identified in both sheep and goat meat [60]. Since then, several studies have shown that these fatty acids negatively affect the taste of goat meat [60,91,94].

Thus, there is no way to relegate to the background the importance of branched-chain fatty acids in the formation of the characteristic flavor of ruminant species. With a more critical look, it is possible to clearly identify which research is necessary for a more detailed understanding of branched fatty acid participation formation of goat and sheep species flavor and aroma. To obtain natural aromas, key points to be studied must deal with the production of methylated fatty acids during enzymatic hydrolysis processes.

## 6. Conclusions

Focusing on bettering suitability as a human food source, many efforts have been made to use sheep and goat by-products in diversified products, such as in food preparations or processed meat products, and also to obtain hydrolyzed proteins and functional peptides. However, the use of these by-products in human food is still challenging, especially with regard to consumer acceptance.

The nutritional, technological, functional, bioactive, and potential flavoring aspects of goat and sheep by-products are remarkable. Not using these by-products brings a loss of potential revenue as well as increasing disposal processing costs.

Many factors influence the potential for fortification in foods, such as nutritional value, pre-treatment, nutrient bioavailability, interaction with other ingredients, sensory characteristics, and regulatory requirements. Further, in the use of such by-products, food safety must be the priority.

Extraction and isolation of bioactive compounds from goat and sheep by-products (antimicrobial, antioxidant, and antibacterial properties) through enzymatic hydrolysis must also be studied in greater depth and might receive special attention as well since their peptides possess many bioactivities. The scarcity of studies on flavor potential, flavoring formulations, and flavor molecules that can be produced from these by-products is noted.

In view of the considerations above, it is evident that the use of goat and sheep by-products in foods will make a significant contribution to the world economy, an excellent solution for strengthening animal production while helping to initiate zero waste management policies.

## Figures and Tables

**Figure 1 animals-12-03277-f001:**
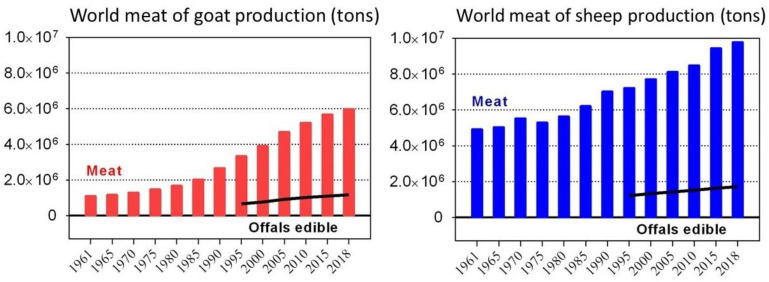
World production of edible meat and meat by-products.

**Table 1 animals-12-03277-t001:** Nutritional composition of goat meat and edible by-products.

	Meat	Brain	Heart	Kidneys	Liver	Lung	Gut	Spleen	Tongue
Centesimal Composition (g/100 g), Total Caloric Value—VCT (kcal/100 g) and Cholesterol (mg/100 g)									
Moisture	71.2–78	76.74	75.15–80.4	80.26	73.37–74.0	79.03–80.3	80.94	77.93	68.4
Ash	0.88–1.1	1.29	0.95	1.1	1.3	0.74	0.23	1.15	1.02
Proteins	18.7–23.1	13.82	16.76–9.38	15.6	20.32–2.63	16.86–7.19	15.36	18.45	16.63
Fat	1.8–6.4	8.06	4.40–6.36	2.93–1.21	4.88–5.98	2.26–3.09	3.36	2.37	13.68
Total Caloric Value (TCV)	91–150	127	118	89	126	96	92	96	189
Cholesterol	45–75	1340	122	419	415	448	113	243	206
Essential amino acids (EAA–g/100 g protein)									
Tryptophan	0.0033	1.36	1.31	1.5	1.22	1.45	1.34	1.46	1.45
Histidine	0.046	5.29	5.38	5.23	4.23	4.64	4.4	4.62	4.7
Threonine	0.046	5.29	5.43	5.69	6.58	5,74	4.16	6.27	4.67
Valine	0.043	4.7	5.08	5.44	4.53	5.15	5.24	5.27	4.5
Methionine	-	6.01	5.59	5.52	6.27	6.07	5.55	6.49	5.7
Isoleucine	0.027	2.98	3.08	3.41	3.04	2.54	3.59	2.7	3.18
Leucine	0.061	5.21	6.35	4.64	6.02	5.82	5.99	5.5	5.59
Phenylalanine	0.028	6.61	6.25	5,74	5.64	7.16	7.76	6.58	7.02
Tyrosine	0.021	4.68	3.97	4.77	3.5	4.51	4.11	3.74	3.97
Lysine	0.058	8.24	7.04	8.48	8.39	8.04	7.22	8.89	7.05
Non-essential amino acids (NEAA—g/100 g protein)									
Aspartic acid	0.024	5.27	1.29	5.29	4.2	3.05	13.68	4.82	8.47
Glutamic acid	0.068	6.71	1.71	4.74	4.38	2.41	13.14	4.27	8.15
Serine	0.022	6.98	1.71	5.08	4.73	2.66	12.62	4.08	8.43
Glycine	0.151	5.29	1.71	4.93	4.76	2.49	12.09	4.61	8.74
Alanine	0.230	6.06	1.48	4.58	5.59	2.46	11.41	3.26	7.77
Arginine	-	8.2	1.62	4.48	5.02	2.88	11.46	3.24	8.33
Proline	0.038	7.86	1.37	4.26	4.54	2.48	12.26	3.44	8.8
Cysteine	0.0003	6.34	1.45	4.64	4.78	2.38	13.47	4.15	8.26
EAA + NEAA	0.866	5.27	1.29	5.29	4.2	3.05	13.68	4.82	8.47
Saturated Fatty Acids (g/100 g)									
12:00	-	0.03–0.25	0	0.01–0.04	0.01	0.04	0.02–0.03	0.02	0.01
14:00	0.0488	0.41–0.48	0.03	0.13–0.23	0.07	0.04–0.07	0.03–0.09	0.15	0.06
16:00	0.6098	2.53–3.29	1	0.76–1.10	0.56	0.74–0.97	0.43–0.91	0.83	0.55
18:00	0.7561	2.40–2.52	1.01	1.78	0.47	0.94–1.10	0.32–0.51	0.67	0.55
20:00	0.0439	0	0.03	0	0	0	0.01	0	0.01
22:00	0.0244	0	0.03	0	0	0	0	0	0
24:00	-	0	0.04	0.01	0.03	0.05	0.02	0.01	0.02
SFA	1.483	5.59–6.63	2.14	1.93–3.29	1.14	1.77–2.23	0.88–1.56	1.68	1,2
Mono Unsaturated Fatty Acids (g/100 g)									
16:1	0.0244	0.43–0.62	0.02	0.04–0.15	0.02	0.04–0.07	0.03–0.04	0.06	0.02
18:1	0.9024	5.00–7.82	1.31	0.95–1.24	0.52	0.88–0.95	0.40–0.84	1.06	0.58
20:1	-	0.26	0.12	0.03–0.05	0.02	0.07	0.01	0.03	0.02
MUFAs	0.9068	5.88–8.46	1.45	1.02–1.77	0.56	0.92–1.40	0.60–0.89	1.15	0.62
Polyunsaturated Fatty Acids (g/100 g)									
18:2	0.0537	0.40–0.48	0.04	0.29–0.43	0.24	0.32–0.37	0.08–0.13	0.15	0.14
18:3	0.0195	0.47–0.51	0.08	0.01	0.01	0.01–0.06	0.02	0.01	0.01
20:4	-	0.06–0.07	0.3	0.01–0.13	0.23	0.36	0.02–0.22	0.08	0.2
22:6 n-3	0.0061	0	0	0.52	0	0	0.14	0	0
PUFAs	0.0793	0.94–1.06	0.94	0.43–0.53	0.48	0.55–0.93	0.13–0.37	0.24	0.35
Minerals (mg/100 g)									
Al	Na	0.22	0.37	0.19	0.53	0.32	0.33	0.29	0.4
Ar	Na	0.56	0.93	1.44	1.49	1.18	0.18	1.18	0.57
Ha	15.5–17.8	10.61	5,71	12.37	6.09	10.44	20.87	8.35	7.11
Cr	Na	0.01	0.01	0.01	0.01	0.04	0.01	0.02	0.01
Cu	0.07–0.23	0.3	0.34	0.25	3.7	0.22	0.07	0.13	0.16
Fe	2.0–20.84	1.63	3.38	6.73	6.56	7.1	0.81	51.41	2.09
Mg	20.4–28.9	13.94	18	18.6	18	11.48	13	15.3	19.39
Mn	0.007–0.021	0.04	0.03	0.08	0.26	0.04	0.07	0.03	0.04
Hg	Na	0.03	0.04	0.38	0.44	0.07	0.15	0.05	0.2
Mo	Na	0	0.07	0.12	0.12	0.02	0.03	0.09	0
Ni	Na	0	0	0	0	0	0	0	0
P	212–275	296	174	220	336	209	59.2	266	212
K	301–413	296	224	195	284	180	45	368	223
Se	8.8	21.7	15	142	48.12	26	15.59	48.55	22.76
Na	98.5–858.3	132	73.21	184	55.41	85.72	22.38	52.4	118
Zn	2.0–3.59	1.08	1.45	1.73	3.48	1.7	1.66	1.95	2.44
Water-soluble vitamins (mg/100 g)									
B1-Thiamin	0.11	0.13	0.17	2.04	1.52	0.28	1.3	0.13	0.17
B2-Riboflavin	0.49	0.14	0.22	3.04	0.88	0.17	2.54	0.14	0.22
B3-Niacin	-	0.08	0.32	2.79	0.83	0.14	1.8	0.08	0.32
Pantothenic acid	-	0.36	0.33	5.97	1.72	0.23	1.63	0.36	0.33
B6-Pyridoxine	-	0.2	0.37	12.88	7.01	0.65	178	0.2	0.37
B9-Folic acid	-	0.02	0.1	0.81	0.21	0.2	2.36	0.02	0.1
B12-Cyanocobalamin	1.13								
Vit C	-								
Fat-soluble vitamins (mg/100 g)									
Vit. A	-	0.002	0.006	0.069	15.66	0.003	0.002	0.008	0.009
Vit. E	-	0.07	0.09	0.09	0.61	0.08	0.10	0.08	0.08
Vit. D	-	-	-	-	-	-	-	-	-
Vit. K	-	0.0038	0.011	0.0035	0.013	0.0046	0.012	0.009	0.002

Sources: [4,39,40].

**Table 2 animals-12-03277-t002:** Nutritional composition of sheep meat and edible by-products.

	Beef	Brain	Heart	Kidneys	Liver	Lung	Gut	Spleen	Tongue
Centesimal Composition (g/100 g), Total Caloric Value—VCT (kcal/100 g) and Cholesterol (mg/100 g)									
Moisture	74.05	78.36–79.2	76.7–77.06	79.23–79.77	69.71–71.37	79.7–80.41	78.81	78.15–79.66	66.6–68.77
Ash	1.15	1.19–1.33	0.93–0.97	1.00–1.26	1.26–1.44	0.97–1.10	0.26	1.17–1.3	0.92–1.06
Proteins	24	10.4–13.05	16.47–8.19	15.74–6.22	20.38–2.26	16.7–16.12	16.76	16.02–17.2	15.7–16.61
Fat	8.1	7.29–8.58	3.66–5.68	2.92–2.95	4.87–5.02	2.41–2.6	4.05	3.05–3.1	13.53–17.17
Total Caloric Value (TCV)	175	118–122	106–122	92–97	134–139	87–95	104	92	187–222
Cholesterol	66	1336–1352	112–135	299–337	371–430	431	113	250–262	156–210
Essential amino acids (g/100 g protein)									
Tryptophan	0.073	1.29	1.4	1.59	1.19	1.36	1.46	1.38	1.36
Histidine	0.198	5.54	5.93	5.08	4.39	4.56	4.72	5.05	5,71
Threonine	0.267	5.3	5.45	4.89	5.52	5.41	3.86	6.94	4.87
Valine	0.337	4.19	4.86	5.11	4.68	4.53	6.05	5.5	3.62
Methionine	0.160	6.04	5.56	5.04	5.12	6.63	5.33	7.04	5.22
Isoleucine	0.302	2.47	3.18	3.78	2.83	2.58	3.44	2.89	3.64
Leucine	0.487	5.21	6.77	6.34	6.2	5.02	5.78	5.9	5.45
Phenylalanine	0.253	6.83	6.45	5.61	5.66	7.43	7.17	6.55	6.79
Tyrosine	0.210	4.23	3.81	4.36	3.67	3.37	4.42	3.34	3.78
Lysine	0.552	8.51	7.86	7.69	7.79	7.78	7.05	8.91	7.49
Non-essential amino acids (g/100 g protein)									
Aspartic acid	0.550	5.28	5.5	6.65	5.17	8.33	8.76	6.76	6.28
Glutamic acid	0.907	1.34	1.61	1.69	1.68	1.4	1.48	1.46	1.48
Serine	0.232	5.19	4.72	4.91	4.91	4.64	4.36	4.39	4.79
Glycine	0.305	3.89	4.23	4.9	4.79	5.48	5.11	4.38	4.3
Alanine	0.376	3.2	2.09	2.57	2.64	2.48	2.6	2.27	2.72
Arginine	0.371	13.47	13.75	12.78	10.93	10.53	10.8	12.11	14.42
Proline	0.262	4.69	4.15	3.78	4.53	3.22	3.32	4.02	3.72
Cysteine	0.075	8.08	7.65	8.82	8.22	7.27	8.65	8.83	8.3
EAA + NEAA	5.917	94.75	94.97	95.59	89.92	92.02	94.36	97.72	93.94
Saturated Fatty Acids (g/100 g)									
12:00	0.01	0.00	0.02	0.00–0.01	0.00	0.01	0.01	0.01	0.03–0.25
14:00	0.15	0.02–0.04	0.13–0.20	0.03–0.06	0.02–0.05	0.04	0.15	0.07	0.41–0.48
4:00	1.01	0.94–1.06	0.77–0.86	0.42–0.51	0.68	0.65	1.00	0.66	2.53–3.29
18:00	0.643	1.00–1.07	0.69–1.16	0.52–0.55	1.08–1.12	0.44	0.80	0.75	2.40–2.52
20:00	-	0.02	0.00	0.00	0.00	0.00	0.00	0.01	0.00
22:00	-	0.01	0.00	0.00	0.00	0.00	0.00	0.00	0.00
24:00	-	0.05	0.00	0.00	0.04	0.00	0.00	0.03	0.00
SFA	3.35	2.04–2.19	1.68–2.25	1–1.12	1.82–1.94	0.89–1.14	1.96	1.03–1.53	5.59–6.63
Mono Unsaturated Fatty Acids (g/100 g)									
16:1	0.144	0.03–0.05	0.03–0.07	0.02–0.04	0.03–0.13	0.02	0.08	0.02	0.43–0.62
18:1	2.21	1.06–1.29	0.82–1.38	0.54–0.55	0.74–0.92	0.73	1.42	0.72	5.00–7.82
20:1	-	0.09	0.02–0.04	0.01–0.03	0.00	0.02	0.03	0.02	0.26
MUFAs	2.37	1.18–1.55	0.87–1.6	0.59–0.63	0.77–1.05	0.67–0.77	1.53	0.76–0.81	5.88–8.46
Polyunsaturated Fatty Acids (g/100 g)									
18:2	0.33	0.03–0.04	0.21–0.24	0.21–0.22	0.32–0.35	0.12	0.13	0.18	0.40–0.48
18:3	0.04	0.08	0.03–0.13	0.01–0.07	0.07	0.01	0.01	0.01	0.47–0.51
20:4	0.06	0.23–0.30	0.08–0.09	0.14–0.23	0.36	0.17	0.07	0.28	0.06–0.07
22:6 n-3	-	0.45–0.49	0.01–0.03	0.00–0.03	0.23	0.00	0.00	0.00	0.00
PUFAs	0.42	0.87–0.88	0.33–0.55	0.46–0.55	0.75–1.01	0.30–0.35	0.21	0.23–0.47	0.94–1.06
Minerals (mg/100 g)									
Al	Na	0.14	0.03	0.19	0.16	0.48	0.27	0.16	0.28
Ar	Na	0.73	6.04	2.1	1.66	1.85	0.1	0.92	0.34
Ca	5–7	9.0–10.27	6.0–6.79	13.0–14.15	5.92–7.0	7.74–10.0	8.0–14.85	9.0–11.49	8.05–9.0
Cr	Na	0	0.01	0.01	0.03	0.02	0.02	0.01	0.08
Cu	0.120	0.24–0.28	0.29–0.40	0.38–0.45	5.70–6.98	0.25–0.41	0.06	0.12–0.16	0.17–0.21
Fe	1.0–2.2	1.75–1.93	3.07–4.60	6.11–6.38	6.15–7.37	6.40–8.58	0.90–2.30	41.89–53.11	2.46–2.65
Mg	24	12.0–14.08	15.19–17.0	17.0–17.46	17.91–19.0	11.68–14.0	17.81–21.0	17.23–21.0	21.0–24.53
Mn	0.010	0.04	0.03–0.05	0.11–0.69	0.18–0.28	0.02–0.03	0.04–0.16	0.04–0.05	0.03–0.05
Hg	Na	0.02	0.03	0.61	0.16	0.07	0.19	0.04	0.45
Mo	Na	0	0	0.02	0.15	0.01	0	0.04	0
Ni	Na	0.01	0.01	0.01	0.02	0.01	0.01	0.01	0.02
P	176–215	270.0–271.0	163.0–175.0	204.0–246.0	334.0–364.0	187.0–219.0	55.61–400.0	266.0–280.0	184.0–207.0
K	333	296.0–312.0	225.0–316.0	198.0–277.0	280.0–313.0	204.0–238.0	42.0–48.75	327.0–358.0	220.0–257.0
Se	Na	34.6	33.3	127	43.38	16.12	13.92	64.53	23.2
Na	72	112.0–122.0	72.62–89.0	156.0–163.0	55.82–70.0	109.0–157.0	18.37–75.0	50.58–84.0	78.0–185.0
Zn	3.83	1.11–1.17	1.31–1.87	1.92–2.24	3.73–4.66	1.58–1.80	1.89–1.93	1.71–2.84	2.32–2.46
Water-soluble vitamins (mg/100 g)									
B1-Thiamin	0.102	0.12	0.15	0.07	0.34	0.17	0.02	0.07	0.35
B2-Riboflavin	0.267	0.21	0.2	0.23	0.3	0.35	0.08	0.23	0.31
B3-Niacin	6.29	2.66	2.78	2.75	5.43	15.66	0.63	5.42	5.51
Pantothenic acid	0.685	1.73	0.35	0.88	1.66	6.95	0.2	6.07	6.22
B6- Pyridoxine	0.15	0.33	0.16	0.02	0.3	0.26	0.22	0.27	0.51
B9- Folic acid	0.023	1.86	2.4	13.89	2.17	206	1.87	3.19	46.21
B12–Cyanocobalamin	0.0026								
Vit. C	1.0								
Fat-soluble vitamins (mg/100 g)									
Vit. A	0.045	0.0015	0.0047	0.067	14,106	0.0024	0.003	0.0066	0.0073
Vit. E	-	0.09	0.15	0.10	0.60	0.08	0.20	0.10	0.07
Vit. D	-								
Vit. K	-	0.0028	0.0097	0.0029	0.0147	0.005	0.011	0.0075	0.0015

Sources: [4,37,40].

**Table 3 animals-12-03277-t003:** Goat and sheep products processed from by-products.

Goat/Lamb Ethnic Meats/Foods			
Name	Origin	Description	Reference
Blomor	Iceland	Sausage-type (lamb’s blood, canned and fermented)	[9]
Lifrarpylsa	Iceland	Sausage-type (lamb’s blood, canned and fermented)	[9]
Osbana	Algeria, Tunisia, Libya and Morocco	Sausage type (lamb by-products-heart, liver, spleen and kidneys)	[8]
Kourdass	North Africa, Mediterranean	Sausage type lamb by-products-stomach, intestines, liver, lung, spleen and fat.	[8]
Geema	Kumano Himalayas/Northeast India	Sausage type (minced goat meat, fresh blood, placed in the small intestine of goat).	[60]
Air-dried	Kumano Himalayas/Northeast India	Sausage type (minced goat meat, goat lungs, stuffed in goat’s large intestines).	[61]
Mcharmla	Algeria and Morocco	Lamb dish prepared with chopped liver, spices, cooked 20/30 min.	[8]
Goat Buchada	Northeast of Brazil	Goat dish prepared with minced by-products (heart, lungs, liver, intestine, blood, spices) stuffed in goat/lamb stomach, sewn, and cooked.	[50,53,62,63]
Sheep Buchada	blood, viscera	Lamb dish prepared with minced by-products (heart, lungs, liver, intestine, blood, spices) stuffed in goat/lamb stomach, sewn, and cooked.	[64]
Sarapatel	Northeast of Brazil	Goat and lamb dish prepared with chopped offal (heart, lungs, liver, intestine, blood, spices) and cooked.	[48]
Tarfa-gara	Algeria	Goat and lamb dish prepared with minced by-products (heart, lungs, liver, intestine, blood, spices) and cooked.	[8]
Klaya	Tunisia	Ready-to-eat lamb product, prepared with meat and offal (liver, kidney, fat, seasoning) boiled and fried in oil.	[8]
Hutspungar	Iceland	Lamb dish prepared with pressed testicles	[9]
Kheuri	Sikkim India	Stuffed yak or beef mixture pressed into the sheep’s stomach and hung outdoors for 1–2 months.	[10]
Goat/lamb offal-meat products			
Smoked goat sausage	Blood, viscera	[16,17]	
Goat chorizo	Heart, kidneys	[18]	
Goat pate	Blood, liver	[11]	
Sheep pate	Blood, liver, meat traces	[12,13]	
Goat and sheep pate	Liver	[14]	
Goat and sheep pate	Liver	[15]	

**Table 4 animals-12-03277-t004:** Hydrolyzed proteins from slaughter by-products of goats and sheep.

Products	By-Products	Reference
Flavoring/flavor enhancer	Liver, heart and lung	[7]
Natural antioxidant	Stomach and intestine	[70]
Nutritional supplement	Stomach, large and small intestines	[71]
Food additive Additive and natural antioxidant	Blood plasma Viscera	[72] [73]
Natural antioxidant	Abomasum	[74]
Collagen	Bone, cartilage, carcass and meat trimmings	[75]
Flavoring	Bone	[21]

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
