# Peer review of "Valuation of Goat and Sheep By-Products: Challenges and Opportunities for Their Use"

_animals, 2022, doi:10.3390/ani12233277_

Round 1
Reviewer 1 Report
Corrections have been marked in the PDF.

Reviewer 2 Report
Valuation of goat and sheep by-products: challenges and opportunities for their use
This review comprehensively analyses the nutritional quality of the edible by-products of the slaughter of small ruminants. It also presents notes to promote and encourage their use, production issues and the challenges and concerns that using these by-products requires.
This review contributes to deepening scientific knowledge about using goat and sheep by-products.
The review is well-organized and simple to follow. Furthermore, it is supported by relevant articles on the subject. The text approaches the subject with scientific accuracy and rigor. The Tables are relevant for understanding the article. Tables need to be more uniform, although it is understood that there may be constraints due to their size. Finally, the conclusions are appropriate to the text developed.
Some detailed comments are below:
L64 impacts, through enabling production “change with” impacts by enabling the production
L82 This review, through a broad scientific approach, will contribute to Through a broad scientific approach, this review will contribute to
L84 knowledge which has “change with” knowledge that has
L86 sheep products has seen “change with” sheep products have seen
Figure 1 Please improve the quality and do your best to make the figures the same dimension
L112 as by-product which then “change with” as a by-product which then
L156 ruminants possess all of “change with” ruminants possesses all of
L188 itself they enjoy comparable “change with” itself, they enjoy comparable
L205 spleen, however the “change with” spleen, however, the
L220 In Tables 1 and 2 above, are compiled in detail, data “change with” Tables 1 and 2 are compiled in detail data
L226 It is interesting to note that the highest “change with” Interestingly, the highest
L245 amounts of carbohydrate “change with” amounts of carbohydrates
L273 presents little variation in values, essential and “change with” presents slight variation in values; essential and
L324 by-product, mostly contains “change with” by-product, mainly contains
L390 Thiamine is therefore a “change with” Thiamine is a
L402 Southeast Asia, and Africa, demand is low in the US, yet “change with” Southeast Asia and Africa, and demand is low in the US yet
L496 preparation has revealed “change with” preparation have revealed
L508 The various results obtained in the aforementioned studies serve “change with” The different results obtained in the studies mentioned above serve
L675 The results revealed that the degree of hydrolysis is an important indicator in the preparation of these flavorings, with demonstrated great effect on sensory attributes (roasted sheep). “change with” The results revealed that the degree of hydrolysis is an important indicator in preparing these flavorings, with a significant effect on sensory attributes (roasted sheep).
L678 It was also possible to detect 36 key compounds that represent the flavor of mutton. “change with” It was also possible to detect 36 key compounds representing the mutton flavour.
L710 shown that the presence of these fatty acids negatively affect the taste “change with” Since then, several studies have shown that these fatty acids negatively affect the taste
L727 by-products is remarkable “change with” by-products are remarkable
